# In Vitro and In Silico Studies of Human Tyrosyl-DNA Phosphodiesterase 1 (Tdp1) Inhibition by Stereoisomeric Forms of Lipophilic Nucleosides: The Role of Carbohydrate Stereochemistry in Ligand-Enzyme Interactions

**DOI:** 10.3390/molecules27082433

**Published:** 2022-04-09

**Authors:** Nadezhda S. Dyrkheeva, Irina A. Chernyshova, Georgy A. Ivanov, Yuri B. Porozov, Anastasia A. Zenchenko, Vladimir E. Oslovsky, Alexandra L. Zakharenko, Darina I. Nasyrova, Galina N. Likhatskaya, Sergey N. Mikhailov, Olga I. Lavrik, Mikhail S. Drenichev

**Affiliations:** 1Institute of Chemical Biology and Fundamental Medicine, Siberian Branch of the Russian Academy of Sciences, Lavrentiev Ave. 8, 630090 Novosibirsk, Russia; elpida80@mail.ru (N.S.D.); chernyshova0305@gmail.com (I.A.C.); sashaz@niboch.nsc.ru (A.L.Z.); 2Engelhardt Institute of Molecular Biology, Russian Academy of Sciences, Vavilova Str. 32, 119991 Moscow, Russia; georgyivanovk423@gmail.com (G.A.I.); yuri.porozov@gmail.com (Y.B.P.); kolomatchenkoa@yandex.ru (A.A.Z.); vladimiroslovsky@gmail.com (V.E.O.); smikh@eimb.ru (S.N.M.); 3World-Class Research Center “Digital Biodesign and Personalized Healthcare”, I.M. Sechenov First Moscow State Medical University, Trubetskaya Str. 8/2, 119991 Moscow, Russia; 4Department of Computational Biology, Sirius University of Science and Technology, Olympic Ave. 1, 354340 Sochi, Russia; 5Department of Physical and Chemical Biology and Biotechnology, Altai State University, Pr. Lenina 61, 656049 Barnaul, Russia; 6Faculty of Chemistry, National Research University Higher School of Economics, Pokrovsky Boulevard 11-10, 101000 Moscow, Russia; nasyrova-01@mail.ru; 7Laboratory of Bioassays and Mechanism of Action of Biologically Active Compounds, G.B. Elyakov Pacific Institute of Bioorganic Chemistry, Far Eastern Branch of Russian Academy of Sciences, Prospekt 100 Let Vladivostoky 159, 690022 Vladivostok, Russia; galin56@mail.ru

**Keywords:** nucleosides, pentafuranose, DNA repair, tyrosyl-DNA phosphodiesterase 1, Tdp1 inhibition, chirality, stereoisomers

## Abstract

Inhibition of human DNA repair enzyme tyrosyl-DNA phosphodiesterase 1 (Tdp1) by different chiral lipophilic nucleoside derivatives was studied. New Tdp1 inhibitors were found in the series of the studied compounds with IC_50_ = 2.7–6.7 μM. It was shown that D-lipophilic nucleoside derivatives manifested higher inhibition activity than their L-analogs, and configuration of the carbohydrate moiety can influence the mechanism of Tdp1 inhibition.

## 1. Introduction

Chirality plays an important role in the development of novel drugs, improvement of their selectivity, and characterization of receptors [1,2]. Most biopolymers comprised of L-amino acids (proteins) and/or D-carbohydrates (nucleic acids, oligosaccharides, and glycolipids) are chiral. Therefore, enantiomerically pure drugs can strengthen the interactions with a specific target, receptor or enzyme due to spatial accessibility to complementary sites on the target. For example, the D-threo (1*R*, 2*R*) isomer of chloramphenicol (1-(4-nitrophenyl)-2-dichloroacetylaminopropan-1,3-diol) is used in medical practice for the treatment of infections caused by Gram-positive and Gram-negative bacteria and acts by binding to the 70S ribosome and thus inhibiting the peptidyl transferase reaction, whereas the L-erythro (1*R*, 2*S*), D-erythro (1*S*, 2*R*), and L-threo (1*S*, 2*S*) isomers are all inactive as antibacterial agents [1]. *R*-thalidomide is a potent sedative drug while S-thalidomide can cause adverse effects [3]. Various effects of stereoisomers of different compounds on human taste buds (lyoniresinol derivatives, 4-mercapto-2-hexanone, 4-acetylthio-2-hexanone) have also been described [3].

Since the 1980s, more than 450 chiral drugs have been developed from natural and synthetic products. Even though the proportion of synthetic drugs in the form of single isomers has increased over the past decades, mixtures of stereoisomers are found in the majority of therapeutic groups [1]. Recent investigations have developed novel enantiomeric drugs based on small molecules and supramolecular complexes. Among the low molecular weight compounds, there are the following examples: the selective quinone methide triterpene inhibitor of heat shock protein 90 (Hsp90); vanilloid receptor antagonists for the treatment of chronic pain based on the natural products capsaicin and resiniferatoxin; (S)-crizotinib, a nanomolar kinase inhibitor of the nucleotide pool sanitizing enzyme MTH1, that is used for the treatment of cancer; the diastereomeric selective binding of monofunctional phenanthriplatin to DNA; simultaneous binding of enantiomeric phenazine biosynthetic intermediates to an enzyme active site [4,5]. Among the supramolecular chiral compounds, stereoselective DNA binding by ferric metallosupramolecular complexes with antibiotic and anticancer activity have been developed [4].

The design of novel drugs based on nucleoside structure is one of the most fruitful areas of medicinal chemistry. Approximately 100 drugs were developed based on nucleosides, half of them being antiviral, and a quarter of them being antitumor drugs [6,7,8,9]. Nucleoside moieties are the components of nucleic acids and coenzymes [10]. Several enantiomeric pure inhibitors of protein kinases were proposed on the basis of carbocyclic adenine derivatives [11]. Enantiomerically pure (1′R,4′R)-dioxolane thymine (D-erythro-form) exhibited a potent anti-HIV activity due to its rapid phosphorylation in cells [12]. It was shown that stereoisomers of nucleoside phosphorothioates had various effects on nucleotide metabolizing enzymes [13]. A series of L-nucleoside derivatives of lamivudine and telbivudine are used in medicinal practice as antivirals. The mechanism of antiviral activity of L-nucleosides seems to be similar to their D-isomers, but L-nucleoside derivatives are biochemically more stable because of their low propensity to metabolic depletion by cellular enzymes [6,14].

A DNA repair enzyme, tyrosyl-DNA phosphodiesterase 1 (Tdp1), represents an important target for the development of novel antitumor therapeutics based on various scaffolds [15,16,17,18], including nucleosides [19,20]. Tdp1 plays a key role in the removal of Top1-DNA adducts stabilized by Top1 inhibitors such as camptothecin and its clinical analogs. Therefore, Tdp1 inhibition can significantly strengthen the action of clinically used Top1 inhibitors, and thus, increase the efficiency of antitumor therapy [17,18,21]. Diverse Tdp1 inhibitors with IC_50_ values in the range 0.015–10 µM have been described [16,17,18,19,20,22,23,24,25,26,27,28,29,30,31]. Recently, a novel group of efficient low-cytotoxic Tdp1 inhibitors based on lipophilic disaccharide nucleoside derivatives with IC_50_ values in the submicromolar range have been found [19]. It was further shown that the 2ʹ,3ʹ,5′-tri-*O*-benzoylpentafuranose residue is essential for Tdp1 inhibition [20]. This study continues on from our previous findings to investigate Tdp1 inhibition by nucleoside derivatives with native and modified configurations of a carbohydrate moiety.

## 2. Results

### 2.1. Chemical Synthesis

All of the compounds presented in Table 1 were obtained by glycosylation of a heterocyclic moiety or *O*-benzoyl-protection of a nucleoside or a carbohydrate by the previously described procedures (see Appendix A for more information).

### 2.2. Tdp1 Inhibition

The measurement of Tdp1 inhibition was performed using a real-time oligonucleotide biosensor based on the capability of Tdp1 to remove fluorophore quenchers from the 3′-end of DNA [31]. The single-stranded Tdp1 substrate was a 16-mer oligonucleotide containing a 5′-FAM (6-carboxyfluorescein) fluorophore donor and a 3′-BHQ1 quencher (black hole quencher 1). When a quencher is removed by Tdp1, FAM fluorescence flares up and can be detected by a fluorimeter. In the presence of an inhibitor, fluorescence intensity decreases. The obtained Tdp1 activity versus inhibitor concentration curves were used to calculate IC_50_ values. The results of Tdp1 inhibition by the nucleoside derivatives are shown in Table 1. We also studied other structurally related compounds, such as optically active benzoylated arabinofuranose and ribofuranose to ascertain the influence of a heterocyclic base on biological activity (Table 1).

According to the data given in Table 1, all compounds with benzoyl groups inhibited the activity of Tdp1 in the micromolar range of concentrations. The inhibitory activity of 1-*O*-Acetyl-2,3,5-tri-*O*-benzoyl-β-D-ribofuranose (**2D**) was slightly higher than the corresponding L-isomer (**2L**). The same trend was observed for the pair **3D**/**3L**. For the **4D**/**4L** pair, the difference between the D- and L-isomers was one order of magnitude (IC_50_ 2.7 ± 0.6 μM and 25 ± 1 μM, respectively). Thus, the configuration at the C-2 carbon center of the carbohydrate moiety can affect the binding affinity and inhibition of the enzyme. The replacement of the acetoxy group (OAc) in position 1 of **2D** and **2L** with a uracil base did not lead to a significant difference in Tdp1 inhibition, though **3D** appeared to be a slightly more potent inhibitor than **3L**. The introduction of a thymine pyrimidine base (Thy) to the 2,3,5-tri-*O*-benzoyl-β-D/L-ribofuranose residue (**4D** and **4L**) led to a significant difference in Tdp1 inhibition by the D- and L-isomers that was missing from the initial ribofuranoses, **2D** and **2L**. At the same time, the inhibition activity of **3D** and **4D** was comparable with the initial D-ribofuranose derivative **2D** and D-arabinofuranose derivative **1D**, respectively. The removal of the benzoyl groups from **3D** and **3L** nucleoside derivatives led to a significant decrease in Tdp1 inhibition by both isomers (IC_50_ > 50 μM for **8D** and **8L**). Inhibition by 1-(2,3,5-Tri-*O*-benzoyl-β-D-ribofuranosyl)thymine **4D** (IC_50_ = 2.7 μM) was comparable with its 6 positional isomer 6-methyl-1-(2,3,5-tri-*O*-benzoyl-β-D-ribofuranosyl)uracil **4′D** (IC_50_ = 3.4 μM). Therefore, the presence of a methyl group in the heterocyclic moiety (5- and 6-methyl, respectively) slightly influences the activity of the D-ribofuranosyl derivatives and significantly influences the activity of the L-ribofuranosyl derivatives.

Changing the stereochemistry of the methyl substituent influences the difference in the biological activity of D-allo- and L-talo-*C*-methylnucleosides [32,33]. To study the influence of the stereoisomerism of *C*-methyl nucleosides on Tdp1 inhibition, 6-deoxy-D-allofuranosylnucleosides and 6-deoxy-L-talofuranosylnucleosides of both purine and pyrimidine were obtained. The 6-Deoxy-D-allofuranosylnucleosides correspond to 5′(*R*)-*C*-methylribonucleosides **5R**-**7R** and the 6-deoxy-L-talofuranosylnucleosides correspond to 5′(*S*)-*C*-methylribonucleosides **5S**-**7S** (see Table 1).

According to the data given in Table 1, each pair of *R*- and *S*-isomers of benzoylated pyrimidine 5′-*C*-methylribonucleosides **5** and **6** possessed comparable activities. The IC_50_ values for the benzoylated uracil derivatives **5R** and **5S** were comparable with the values of benzoylated D-ribothymidine **4D** and D-uridine **3D**. The benzoylated 5′-*C*-methylcytidine derivatives **6R** and **6S** were two-fold less active than nucleosides **5R** and **5S,** and their inhibition was similar to that of the benzoylated L-carbohydrates **1L** and **2L**. The replacement of the pyrimidine with a purine base did not lead to a significant difference in inhibition between the *R*- and *S*-stereoisomers. Inhibition of Tdp1 by 5′(*S*)-*C*-methyl-2′,3′,5′-tri-*O*-benzoyl-*N*^6^-benzoyladenosine (**7S**) was comparable with the corresponding *R*-isomer **7R**.

### 2.3. Molecular Docking

For the first time, we identified a binding cavity for nucleoside and carbohydrate series by superposition of their conformers complexed with Tdp1 (Appendix A). It was shown that the nucleoside derivatives occupied a cavity very close to the active center of the enzyme. This cavity is formed by residues Tyr204, Cys205, Asp230, Lys231, Leu255 Ala258, Phe259, and Thr261, which are able to mediate hydrophobic and hydrogen-bonding interactions with a low-molecular-weight ligand [34]. Using this method, we defined a region with minimum binding energy between the inhibitors and the enzyme, and two forms of Tdp1 that represent targets for nucleoside inhibitors were modeled as follows: the enzyme−substrate complex Tdp1-DNA (A) and the apo-form (B) (Figure 1, Appendix A). Model A (Figure 1A) was designed to simulate the interaction with uncompetitive inhibitors, and model B (Figure 1B) was designed to simulate the interaction with competitive inhibitors. Nucleoside derivatives **3D** and **3L** tend to bind to a cavity adjacent to the active site of Tdp1 (Figure 1A). The purine 5′-*C*-methylribofuranose derivatives **7R** and **7S** bind to the active site for DNA, thus preventing the formation of the DNA-Tdp1 complex. These derivatives also can interact with the cavity near the active site analogously to **3D** and **3L** by hydrogen bonding and stacking interactions of the benzoyl groups with amino acid residues (Figure 1, **7R**).

### 2.4. Mechanism of Action

The presence of an inhibitor in the reaction mixture affects the kinetic parameters of the enzymatic reaction. The determination of the enzyme behavior in the presence of inhibitors and the estimation of the kinetic parameters for this process can reveal significant insight into the mechanisms of inhibition. Depending on the binding site, as well as what stage of the catalytic process it occurs, there are four main types of inhibition: competitive (the inhibitor competes with the substrate for binding at the active site), non-competitive (inhibitor binds at the allosteric center), mixed (inhibitor binds both the active and allosteric centers), and uncompetitive (inhibitor binds to the enzyme-substrate complex). The type of inhibition is one of the important factors that determine the pharmacodynamic properties of the compound and, in turn, the possible side effects of therapy.

To confirm the interaction model obtained by molecular docking, the type of inhibition by compounds **3D**, **3L**, **7R**, and **7S** was determined by a steady-state kinetic assay. During the experiment, dependence curves of the enzymatic reaction rate (V) on substrate concentration (S) in the presence of different concentrations of the inhibitors were obtained (Appendix A). The apparent Michaelis constant (K_M_) and the maximum reaction rate (V_max_) were calculated according to the Michaelis–Menten equation. Conclusions about the type of inhibition were made from the characteristics of the dependence of V_max_ and K_M_ on the concentration of the inhibitor.

For compounds **3D** and **3L**, both V_max_ and K_M_ values decreased as the concentration of the inhibitor increased, which is typical for an uncompetitive type of inhibition (Appendix A). Such inhibitors bind only to the substrate−enzyme complex and not the apo-enzyme. This model is consistent with the results of molecular modeling, where it was shown that compounds **3D** and **3L** bind to the enzyme near the DNA binding site (substrate).

In the presence of compound **7S**, K_M_ increased but V_max_ remained nearly constant (Appendix A). This is characteristic of the competitive type of inhibition, i.e., **7S** binds to the active center of Tdp1 and prevents the formation of the enzyme-substrate complex necessary for the catalytic reaction.

The nature of the change in the kinetic parameters of the reaction in the presence of compound **7R** did not fit into any of the classical types of inhibition. In this case (Appendix A), V_max_ decreased over the entire range of inhibitor concentrations, yet the K_M_ value increased after it decreased to a concentration equal to IC_50_ (4.4 μM, Table 1), indicating a complex inhibition mechanism (uncompetitive at low inhibitor concentrations and mixed at high).

To clarify the mechanism of action of this compound, the change in the fluorescence anisotropy during the Tdp1 catalytic reaction was studied. This method is based on the ability of fluorescent molecules to be excited by plane-polarized light when their dipole moment is parallel to the plane of polarization. If such a molecule is unable to rotate in space, then the emitted light will be polarized following excitation. Conversely, if the molecule can rotate, then the emitted light is “depolarized”. The minimum anisotropy value corresponds to FAM-labeled DNA (Appendix A, gray graph). When Tdp1 was added, the anisotropy increased due to the binding of the DNA by the enzyme, which prevents its free rotation (Appendix A, red graph).

In the presence of compound **7S**, the fluorescence anisotropy decreased with an increasing concentration of the inhibitor (Figure 2, left). This means that the inhibitor can bind to the active site of Tdp1 instead of the DNA substrate, thereby reducing the activity of the enzyme by competitive inhibition. This is consistent with the data obtained in the kinetic studies. Hence, the fluorescence anisotropy assay can be used to predict competitive inhibition. In the presence of compound **7R**, a similar decrease in fluorescence anisotropy was observed (Figure 2, right). Thus, a competitive component is present when Tdp1 is inhibited by compound **7R**; however, this process is more complicated and requires further study with other methods.

## 3. Discussion

We have previously shown that disaccharide and monosaccharide nucleosides with benzoyl groups inhibit Tdp1 with an IC_50_ in the submicromolar range of concentrations [19,20]. Th current work focuses on the inhibition of Tdp1 by benzoylated chiral nucleoside derivatives as potent nucleoside drug prototypes.

According to the screening of a library consisting of 17 chemically synthesized nucleoside compounds, new efficient Tdp1 inhibitors were found in a series of stereoisomeric purine and pyrimidine nucleoside derivatives with IC_50_ values in the low micromolar range of concentrations (see Table 1). The D-forms of 2,3,5-tri-*O*-benzoylated ribofuranose **2D** and the fully *O*-benzoylated uridine **3D** manifested slightly higher inhibition than the corresponding L-forms, **2L** and **3L**. The inversion of the configuration at the C-2′ atom of the ribofuranose moiety led to a significant difference in Tdp1 inhibition (Figure 3A). The largest difference in inhibition of Tdp1 was measured between the thymine derivatives **4D** and **4L** and the 6-positional isomer **4′D** and **4L**. This result indicates the necessity of the methyl group in pyrimidine and the D-configuration of the ribofuranose moiety for a nucleoside to possess a higher binding capacity for Tdp1. In the series of 5′-*C*-methylribonucleosides, pyrimidine analogs possessing a (*R*)-configuration at the 5′ carbon atom of the ribofuranosyl moiety manifested slightly lower inhibition of Tdp1 than the 5′(*S*)-isomers (Figure 3B). The replacement of the pyrimidine base with purine did not lead to a significant difference in IC_50_ between the (*R*) and (*S*)-isomers (Figure 3B).

It was shown that Tdp1 inhibition depended upon the lipophilicity of nucleoside derivatives expressed by benzoic acid residues because their removal from **3D** and **3L** nucleoside derivatives led to a significant decrease in Tdp1 inhibition (IC_50_ > 50 μM, Table 1, compounds **8D** and **8L**). According to the calculations for the distribution coefficient (clogP) between 1-octanol-water phases carried out with Instant JChem software, the value of the Tdp1 inhibition effect for the D-nucleoside derivatives **1D**-**4D**, **4′D** and the 5′(*R*)-stereoisomers **5R**-**7R** depends upon the lipophilicity associated with a benzoyl moiety as a part of the modified carbohydrate moiety (Table 1).

The selectivity of the studied nucleosides for Tdp1 can be estimated from the calculation of the eudismic ratio ER (Table 1, column 6) for each pair of stereoisomers. The stereoisomer with the greater inhibitory activity is termed the eutomer (Eut) and that with the lower activity, the distomer (Dis) [1]. ER was calculated as the ratio of IC_50_ value for the eutomer to IC_50_ value for the distomer. Logarithmic values of ER were also used for calculations (Table 1, column 7). Compounds **2D**/**2L**, **5R**/**5S**, and **6R**/**6S** manifested comparable selectivity as their ER values were close to 1 (see Table 1). The arabinofuranose derivative **1D** was a more selective inhibitor than **1L** (ER = 0.343). Isomeric pairs **2D**/**2L** and **3D**/**3L** were characterized by comparable ER values. Thus, the replacement of the α-methoxide group at position C-1 of the D-2,3,5-tri-*O*-benzoylarabinofuranosyl moiety with a uracil base at the β-configuration and simultaneous inversion of the configuration at position C-2 of the D-2,3,5-tri-O-benzoylarabinofuranosyl moiety with the formation of a D-2,3,5-tri-O-benzoylribofuranosyl moiety leads to comparable selectivity for the D- and L-derivatives. The lowest ER value (highest pER) was calculated for the thymine derivatives **4D** and **4L** (Table 1, ER = 0.108, pER close to 1; Appendix A). Thus, carbohydrate stereoisomers or a thymine ribonucleoside can be applied to gain prevailing Tdp1 inhibition by D-stereoisomers over L-stereoisomers.

The experimental results for the mechanism of action agreed with the molecular modeling data for intermolecular enzyme-ligand interactions. Compounds **3D** and **3L** tended to bind to the cavity near the active center of Tdp1 (uncompetitive inhibition) by a hydrophobic interaction of the phenyl residue from the 5′-*O*-benzoyl residue with Trp590 (Figure 4A,B). A hydrogen intermolecular bond between the 2′-oxygen atom and Gly458 through the associated aqueous molecule participates in the formation of compound **3D** with Tdp1. Compound **3L** forms a hydrogen bond between the 2′-*O*-benzoyl residue and Cys205 (Figure 4B). The uracil base in **3D** associates with Tdp1 by the formation of a hydrogen bond between the C2 carbonyl group of uracil and Tyr204 (Figure 4A, Table 2). The uracil base in **3L** has more possibilities of forming a hydrogen bond with one of several amino acids in the enzyme (Figure 4B). Compound **4D** manifested a similar stacking interaction with Trp590 as **3D** and formed a similar H_2_O-mediated hydrogen bond with Gly458 as **3D**, but also formed one additional hydrogen bond with an aqueous molecule near Asp230 (Appendix A, Table 2). The formation of hydrogen bonds and stacking interactions with **4L** did not look the same as **3L**; instead, **4L** associated with Tdp1 cavity through the formation of hydrogen bonds between the C2 carbonyl group of thymine with Cys205 (instead of Cys 205 and 2′-*O*-benzoyl for **3L)**, and Tyr204 associates with the carbonyl group of 2′-*O*-benzoyl residue (Appendix A, Table 2). In the case of Tdp1 complexation with **4L**, the 5′-*O*-benzoyl residue appears to be far away to stack with Trp590. The evaluation of the binding energy for isomeric pairs **3D**/**3L** and **4D**/**4L** based on sorting by docking score showed similar docking score values for compounds **3D** and **3L**, whereas compounds **4D** and **4L** manifested a difference between their docking score values (Table 2). Thus, it can be concluded that introduction of a methyl substituent in a uracil heterocyclic base clearly influences the nucleoside interaction with Tdp1. The differences in binding energies for **4D** and **4L** can be hypothetically explained by a peculiarity in the calculation of the docking score index, which in terms of thermodynamics, is analogous to the free energy of interaction of an inhibitor with an enzymatic molecule in a crystal structure (characterized by high regularity, which corresponds to a minimal entropy factor ΔS) that is a specific feature of the docking procedure. These conditions differ substantially from inhibition in an aqueous solution, where the interaction of **4D** with the enzyme can be energetically preferable. For example, it may be proposed that the interaction of **4D** with the enzyme strongly increases the entropy factor (ΔS) upon interaction in aqueous solution, and this lowers the system free energy leading to an increase in Tdp1 inhibition by **4D**. Therefore, the docking method allows us to visualize ligand-enzyme interactions and evaluate the binding energy with a docking score index for ligand-enzyme interactions in a crystal structure, which is a potent instrument for various mechanistic studies. The docking score index can also generally indicate differences in inhibition by different stereoisomers in solution, but it cannot reliably evaluate these differences as it does not correlate well with IC_50_ values because of the complex intermolecular interactions in aqueous solutions, which are difficult to predict by computer simulations.

As was shown for the purine 5′-*C*-methylribofuranosyl derivatives, compounds of this type occupied the active site, preventing the formation of the DNA-Tdp1 complex. The 5′ (*S*)-stereoisomer (L-talo) **7S** forms three hydrogen bonds between benzoyl residues and amino acid residues: one between the 2′-*O*-carbonyl group and Thr466, another between the 5′-*O*-carbonyl group and Trp590, and the last between the *N*^6^-carbonyl group and Tyr204 (Appendix A). The 5′(*R*)-stereoisomer (D-allo) **7R** favored the other spatial occupation in the apo-form of Tdp1 due to the formation of four hydrogen bonds between benzoyl residues and amino acid residues: one between the 3′-*O*-carbonyl group and Ser462, two between the 2′-*O*-carbonyl and 5′-*O*-carbonyl groups and associated aqueous molecules, while the hydrogen bond between the *N*^6^-carbonyl group and Tyr204 remains unchanged (Appendix A). The stacking interaction between the adenine base and Trp590 strengthens the binding of the 5′(*R*)-stereoisomer (D-allo) **7R** with Tdp1. When binding with the allosteric site of Tdp1 (Tdp1-DNA), **7R** formed a hydrogen bond with Gly458, mediated by one aqueous molecule and two stacking interactions between Trp590, the phenyl ring of 5′-*O*-benzoyl residue, and the purine moiety (Appendix A). According to our binding energy evaluation (Table 2), **7S** seems to be a more potent competitive inhibitor (docking score −2.506) than **7R** (docking score −1.299), while **7R** tends mostly towards uncompetitive inhibition by interacting with the allosteric site of Tdp1. The formation of several hydrogen bonds and two stacking interactions leads to the lowest docking score for **7R** (−6.490) among the series of studied stereoisomers (Table 2).

The value of Tdp1 inhibition by lipophilic nucleoside derivatives increased with an increase in the quantity of *O*-benzoyl groups attached to a carbohydrate core [20]. Non-protected nucleosides did not display noticeable inhibition of Tdp1 (Table 1); therefore, it can be concluded that introduction of benzoyl groups into a carbohydrate core of a nucleoside is necessary for inhibition of Tdp1. Three benzoyl functionalities and a heterocyclic base attached to a carbohydrate core form a set of general hydrogen bonds, hydrophobic and stacking interactions, which participate in binding the Tdp1 enzyme. Benzoyl functionalities in lipophilic D/L-ribonucleosides and 5′-*C*-methylribonucleosides interact with different sets of amino acid residues, depending on the structure of the carbohydrate core (Figure 4, Appendix A, Table 2). The structure and stereochemistry of the carbohydrate core thus defines the spatial location of the benzoyl groups and the heterocyclic base and can define their interaction with amino acid residues in a specific site of an enzyme, thereby influencing the mechanism of inhibition. *O*-benzoylated pyrimidine nucleosides tend mostly towards an uncompetitive type of inhibition, regardless of the D- or L-configuration of the ribofuranose moiety (**3D** and **3L**). Fully benzoylated purine 5′(*S*)-*C*-methylribonucleoside (L-talo) (**7S**) tends mostly towards competitive inhibition, but the corresponding D-allo-stereoisomer (**7R**) can display mixed inhibition by several possibilities of binding, both to an allosteric site and the active center of Tdp1. These experimental results are in agreement with the computational data and indicate a cooperative role for the benzoyl groups and the nucleic base in different modes of interaction with the enzyme. A mode of interaction is defined by a structure of a carbohydrate moiety that participates as a core for spatial location of benzoyl residues and nucleic bases.

## 4. Materials and Methods

### 4.1. General Chemistry

The solvents and materials were reagent grade and were used without additional purification. Column chromatography was performed on silica gel (Kieselgel 60 Merck (Darmstadt, Germany), 0.063–0.200 mm). TLC was performed on a Alugram SIL G/UV254 (Macherey-Nagel, Düren, Germany) with UV visualization. ^1^H and ^13^C (with complete proton decoupling) NMR spectra were recorded on a Bruker AMX 300 NMR instrument and are given in the Appendix A. ^1^H-NMR-spectra were recorded at 300 MHz and ^13^C-NMR-spectra at 75 MHz. Chemical shifts in ppm were measured relative to the residual solvent signals as internal standards (CDCl_3_: ^1^H, 7.26 ppm, ^13^C, 77.1 ppm; DMSO-*d*6: ^1^H, 2.50 ppm; ^13^C: 39.5 ppm). Spin-spin coupling constants (*J*) are given in Hz. Carbohydrate derivatives **1(D/L)**-**2(D/L)** were obtained according to the procedures described in [35]. D-Uridine (**3D**, **8D**) and D-ribothymidine derivatives (**4D**) were obtained according to the procedures described in [35,36,37,38,39,40,41,42]. The 5′-*C*-methylribonucleosides were obtained according to the procedures described in [41]. Synthetic procedures and NMR spectra of the obtained compounds are given in the Appendix A. The values for the partition coefficient of the compounds between octanol-water phases (clogP) were calculated using Instant JChem software, version 21.15.0 (ChemAxon^®^, Budapest, Hungary).

### 4.2. Real-Time Detection of Tdp1 Activity

Real-time detection of Tdp1 activity was reported in our previous work [31]. This approach involves measuring the fluorescence intensity during the reaction of quencher removal from a fluorophore-quencher coupled to a DNA-oligonucleotide catalyzed by Tdp1 in the presence of inhibitor (the control samples contained 1% DMSO). Reaction mixtures (200 mL) contained Tdp1 buffer (50mM Tris-HCl, pH 8.0, 50mM NaCl, 7mM β-mercaptoethanol), 50 nM biosensor, testable inhibitor, and purified Tdp1 (1.5 nM). The biosensor was a single-stranded oligonucleotide (5′-[FAM] AAC GTC AGGGTC TTC C [BHQ]-3′) containing a fluorophore at the 5′-end (5,6-FAM) and black hole quencher 1 (BHQ) at the 3′-end that was synthesized in the Laboratory of nucleic acid chemistry at the Institute of Chemical Biology and Fundamental Medicine, Novosibirsk, Russia. The reactions were incubated at a constant temperature of 26 °C in a POLARstar OPTIMA fluorimeter (BMG LABTECH, GmbH, Soest, Germany) set to measure fluorescence every 55 s (Ex485/Em520 nm) during the linear phase (from 0 to 8 min). The values for half maximal inhibitory concentration (IC_50_) were determined using a six-point concentration response curve in three independent experiments and calculated using MARS Data Analysis 2.0 (BMG LABTECH, Soest, Germany).

### 4.3. Steady-State Kinetic Analysis of Tdp1 Enzymatic Reaction

To determine the apparent maximum rate of enzymatic reaction (V_max_), Michaelis constant (K_M_), and possible inhibition mechanism, steady-state kinetic experiments were carried out at five fixed concentrations for the substrate with various inhibitor concentrations. The standard reaction mixtures (200 μL) contained 50 nM, 100 nM, 200 nM, 500 nM, or 1000 nM substrate, testable inhibitor, 1.5 nM recombinant human Tdp1, and reaction buffer components (50 mM Tris-HCl, pH 8.0, 50 mM NaCl, 7 mM β-mercaptoethanol). After the addition of the enzyme, the reaction mixtures were incubated at a constant temperature of 26 °C and measured in a POLARstar OPTIMA fluorimeter (BMG LABTECH, GmbH) set to measure fluorescence every 55 s (Ex485/Em520 nm) during the linear phase (0 to 8 min). The initial data (kinetic curves) were obtained in three independent experiments and statistically processed with OriginPro 8.6.0 software.

### 4.4. Molecular Docking

Two Tdp1 models were used in our docking studies, (A) and enzyme−substrate complex (Tdp1-DNA), and (B) an apo-form. Model B was based on the crystal structure 6MJ5 (resolution 1.85 Å) from the Protein Data Bank (https://www.rcsb.org/structure/6MJ5, accessed on 28 June 2021). The structure of model B was prepared for docking using Schrödinger molecular modeling suite (version 2020-4) by Protein Preparation Wizard (PPW, Schrödinger Suite 2020-4, Schrödinger, LLC, New York, NY, USA, 2020) and optimized by the Loop Refinement module in Prime. The quality of the structure was checked using the Protein Health tool and was confirmed by a free energy calculation with OPLS-3ext force field. This made this model appropriate for conducting molecular docking studies. Model A was obtained by the addition of a DNA molecule from crystal structure 5NW9 (PDB ID: https://www.rcsb.org/structure/5NW9, accessed on 28 June 2021) into the active center using a geometry superposition followed by energy minimization of the complex. The calculations were carried out using Schrodinger 2020-4 software. Molecular docking was performed using Glide, and the precision of docking was XP. Ligands were built in a 2D visualizer before modeling, and then optimized using ligand preparation with the generation of conformers. The LigPrep procedure was to keep the chirality as it was specified in 2D Sketcher. A maximum of 22 conformers per ligand was specified in the docking studies. The active site of the enzymatic target was identified, and a grid generation was carried out. The quantity of positions for each conformer when docking with Tdp1-DNA (model A) was 61 for **3D**-**3L**, **4L**, **7R** and 60 for **4L**. The quantity of positions for each conformer when docking with Tdp1 (model B) was 106 for **7R/7S**. The binding energy of ligands was estimated by sorting the docking scores for each position of the conformers in complex with the enzyme. Binding propensity with the enzyme was estimated by sorting glide emodel and Glide score parameters.

## 5. Conclusions

In conclusion, we have studied the inhibition of Tdp1, an important human DNA repair enzyme and a target of anticancer therapy, by different chiral lipophilic nucleoside derivatives and carbohydrate stereoisomers. New efficient Tdp1 inhibitors were found in a series of studied compounds with IC_50_ values in the micromolar range (2.7–6.7 μM). For the series of synthesized compounds, *O*-benzoylated, D-nucleoside, and D-carbohydrate derivatives manifested greater Tdp1 inhibition than their L-analogs. It was shown that *O*-benzoylated D-lipophilic nucleoside derivatives manifested greater inhibition activity than their L-analogs, but no significant difference in Tdp1 inhibition was observed for the *O*-benzoylated 5′(R)-C-methylribonucleosides compared to the 5′(S)-forms.

According to the computer modeling data, nucleosides **3D** and **3L** bind to the cavity adjacent to the Tdp1 active center, as confirmed by kinetic experiments. The purine 5′-*C*-methylribofuranose derivatives **7R** and **7S** bind to the active site for DNA, thus preventing the formation of the DNA-Tdp1 complex. Their ability to compete with DNA for the active site of the enzyme is also supported by kinetic data.

It can be concluded from the obtained results that the stereochemistry of the carbohydrate moiety determines the spatial location of the benzoyl groups and the heterocyclic base, which favors binding to different sites of Tdp1 and thereby affects a mechanism of Tdp1 inhibition.

## 6. Patents

The structures of the described lipophilic nucleosides as Tdp1 inhibitors are protected by Russian Federation Patent RU№ 2748103 C1 (priority from 13.12.2019).

## Figures and Tables

**Figure 1 molecules-27-02433-f001:**
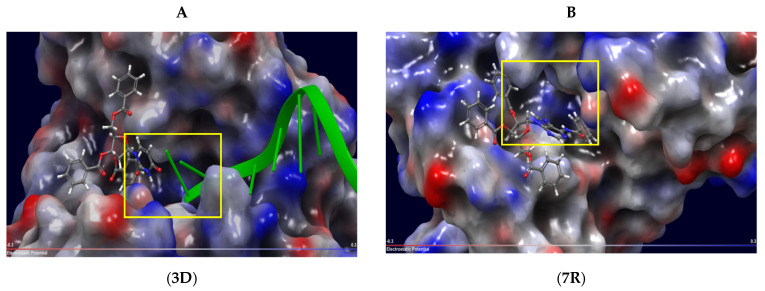
Molecular models of Tdp1 interactions with various types of inhibitors. (**A**) enzyme−substrate complex (Tdp1-DNA); (**B**) apo-form. Negatively charged regions are marked in red, positively charged regions are marked in blue, a DNA fragment is marked in green, and the enzymatic reaction center is marked by a yellow frame.

**Figure 2 molecules-27-02433-f002:**
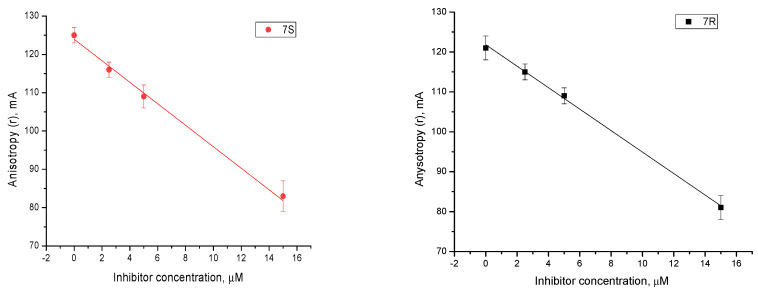
Fluorescence anisotropy assay: a change in the value of fluorescence anisotropy in the presence of compounds **7S** (**left**) and **7R** (**right**).

**Figure 3 molecules-27-02433-f003:**
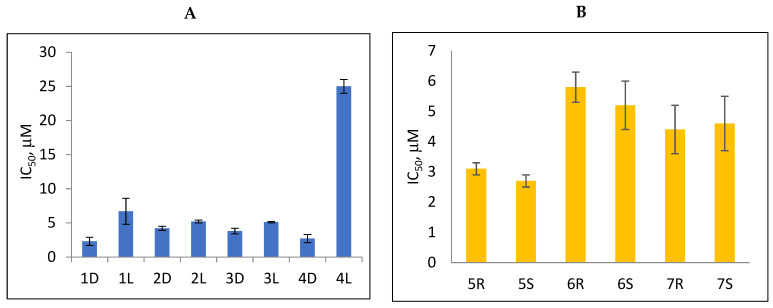
Comparison of Tdp1 inhibition (IC_50_, μM) by (**A**) D- and L- and (**B**) 5′(R)- and 5′(S)-nucleoside derivatives.

**Figure 4 molecules-27-02433-f004:**
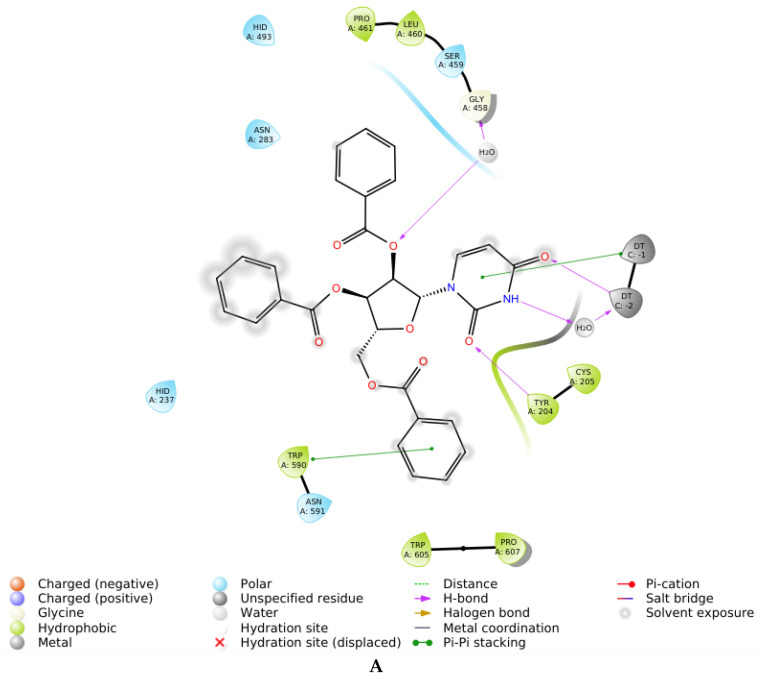
Interaction diagrams for D-nucleoside **3D** (**A**) and L-nucleoside **3L** (**B**).

**Table 1 molecules-27-02433-t001:** Inhibition of Tdp-1 by chiral nucleoside derivatives.

Compound	Structure	clogP ^1^	IC_50_ μM	Eut/Dis	ER ^2^	EI ^3^
**1D**	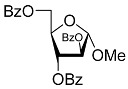	5.83	2.3 ± 0.6	Eut	0.343	0.465
**1L**	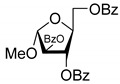	5.83	6.7 ± 1.9	Dis		
**2D**	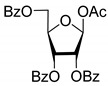	5.62	4.2 ± 0.3	Eut	0.808	0.09
**2L**	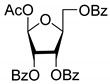	5.62	5.2 ± 0.2	Dis		
**3D**	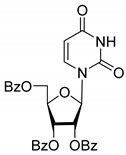	5.07	3.8 ± 0.4	Eut	0.745	0.128
**3L**	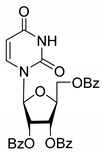	5.07	5.1 ± 0.1	Dis		
**4D**	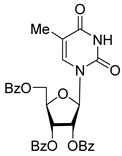	5.47	2.7 ± 0.6	Eut	0.108	0.967
**4L**	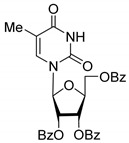	5.47	25 ± 1	Dis		
**4′D**	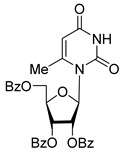	5.27	3.4 ± 0.2	-	-	-
**5R**	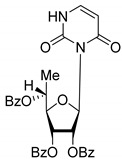	5.49	3.1 ± 0.2	Dis	0.871	0.06
**5S**	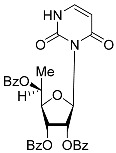	5.49	2.7 ± 0.2	Eut		
**6R**	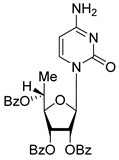	5.10	5.8 ± 0.5	Dis	0.896	0.048
**6S**	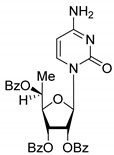	5.10	5.2 ± 0.8	Eut		
**7R**	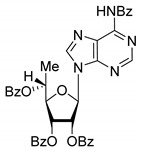	7.73	4.4 ± 0.8	Eut	0.956	0.0193
**7S**	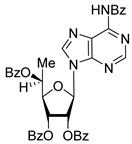	7.73	4.6 ± 0.9	Dis		
**8D**	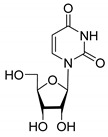	−2.42	>50	-	-	-
**8L**	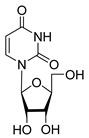	−2.42	>50	-		

^1^ cLogP values (the logarithm of the partition coefficient between n-octanol and water log(C_octanol_/C_water_)) were calculated using Instant JChem software, version 21.15.0 (ChemAxon, https://chemaxon.com, accessed on 9 November 2021). ^2^ ER (eudismic ratio), the ratio of activities of two stereoisomers. ER= IC_50_ (Eutomer)/IC_50_ (Distomer). ^3^ EI = −logER (eudismic index).

**Table 2 molecules-27-02433-t002:** Binding energy evaluation and interactions with amino acid residues for chiral nucleoside Tdp1 inhibitors.

Compound	Inhibition Type	Docking_score ^1^	Interactions with Amino Acids in Enzyme
**3D**	Uncompetitive (Tdp1-DNA)	−5.216	Tyr204-Cys205 (mediated by H_2_O), Gly458 (mediated by H_2_O), Trp590
**3L**	Uncompetitive (Tdp1-DNA)	−5.228	Tyr204, Cys-205, Trp590
**4D**	Uncompetitive (Tdp1-DNA)	−4.670	Gly458 (mediated by H_2_O), Trp590, H_2_O (near Asp230)
**4L**	Uncompetitive (Tdp1-DNA)	−5.362	Tyr204, Cys205, Asn283, H_2_O (near Gly-458)
**7S**	Competitive (Tdp1)	−2.506	Tyr204, Thr466, Trp590
**7R**	Competitive (Tdp1)	−1.299	Tyr204, Ser462, Ser462 (mediated by H_2_O), Trp590
**7R**	Uncompetitive (Tdp1-DNA)	−6.490	Trp590, Gly458 (mediated by H_2_0)

^1^ Average value for 24 optimal conformers *per* ligand (see Appendix A).

## Data Availability

Not applicable.

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
