# Peer review of "In Vitro and In Silico Studies of Human Tyrosyl-DNA Phosphodiesterase 1 (Tdp1) Inhibition by Stereoisomeric Forms of Lipophilic Nucleosides: The Role of Carbohydrate Stereochemistry in Ligand-Enzyme Interactions"

_molecules, 2022, doi:10.3390/molecules27082433_

Round 1

Reviewer 1 Report

Comments on the manuscript ID1624326 by Dyrkheeva et al. entitled “In vitro and in silico studies of human tyrosyl-DNA phosphodiesterase 1 (Tdp1) inhibition by stereoisomeric forms of lipophilic nucleosides. The role of carbohydrate stereochemistry in ligand-enzyme interactions”:

The authors continue their previous studies on Tdp1 inhibition by nucleoside derivatives with native and modified configuration of a carbohydrate moiety. Here they are focused on the role of the carbohydrate stereochemistry in the ligand (nucleoside derivatives) - enzyme (Tdp1) interactions. The article is full of content material and beneficial information, but the information is not presented in a straightforward and understandable manner.

Some shortcomings are listed below:

  • Several different notations are used for the stereochemistry (D/L, R/S, erythro-threo, allo/talo) without explaining the relationship between them (where it exists); stereochemistry of carbohydrates is described mainly by using D/L notation, as opposed to the modern R/S method;
  • There are unexplained abbreviations in the text – FAM, BHQ, etc.;
  • The stereochemistry of the studied chiral nucleotides is not obvious from the chemical structures presented in Table 1;
  • The possibility for different binding of the stereoisomers is not taken into account – the binding cavity for the stereoisomers may differ by one or more residues.

Author Response

1. Several different notations are used for the stereochemistry (D/L, R/S, erythro-threo, allo/talo) without explaining the relationship between them (where it exists); stereochemistry of carbohydrates is described mainly by using D/L notation, as opposed to the modern R/S method;

Answer: we tried to reveal the relationship between different stereochemic notations all over the text, and marked all the changes by green color. As the majority of nucleosides comprise from heterocyclic base and natural carbohydrate D-ribose, the use of carbohydrate D/L-notation for D- and L-nucleosides seems to be not confusing. The use of R/S-notifications upon transition of D-ribose to L-ribose is accompanied by configuration changes at all the stereocenters which must be identified in terms of R/S-notifications, this somehow complicates the resulting name of ribonucleoside. Therefore, we would prefer not to change D- and L-nomenclature of ribofuranose moiety. D-allo- and L-talo-notifications really complicate the text. Therefore, we revealed the name of D-allo- and L-talofuranose in several places (marked by green) in the text with further abbreviated identification of D-allo-form as 5ʹ(R)-stereoisomer an L-talo-form as 5ʹ(S)- stereoisomer to underline stereoisomeric changes at 5ʹ-stereocenter only.

2. There are unexplained abbreviations in the text – FAM, BHQ, etc.;

Answer: corrected. We added the corresponding determinations in the text:

FAM – 6-carboxyfluorescein

BHQ – black hole quencher

3. The stereochemistry of the studied chiral nucleotides is not obvious from the chemical structures presented in Table 1;

Answer: corrected.

4. The possibility for different binding of the stereoisomers is not taken into account – the binding cavity for the stereoisomers may differ by one or more residues.

Answer: Yes, you are right, the binding cavity for stereoisomers differs, which is seen from superposition of carbohydrate and nucleoside derivatives (Fig. S28, Supplementary), where nucleoside molecules bind to enzyme reaction center closer than carbohydrate molecules. The interactions with variable set of amino acids are also confirmed by the data from Table 2. As “ligand-protein” interactions is a very complex process, which is affected by a plenty of physical and chemical parameters, several computer models can describe it. In our work we used computer models which were in accordance with experimental results or at least which did not contradict them (as for compounds 4D/4L in Table 2).

Reviewer 2 Report

A very interesting research and excellent manuscript, Authors have used insilico and invitro analysis to study the differnt 
chiral lipophilic nucleoside derivatives for the inhibition of tyrosyl-DNA phosphodiesterase1 The manuscript is well-written.  
Some Figures need to be improved, since the quality of the figure is not journal acceptable. 

A few comments only

Highlights + Abstract = O.K.

 Introduction
Clearly written 

Materials san Methods
Well-described, with sufficient details to enable other research groups to repeat similar experiments.

Results and discussion

An excellent report of the findings and their explanation. No comments

Conclusions
Somewhat disappointing, since rather descriptive. Readers certainly will appreciate some detailed findings (of section 3). The abstract is better in this regard.

Author Response

1. Some Figures need to be improved, since the quality of the figure is not journal acceptable. 

Answer: we have improved the quality of pictures from figure 1 to more than 3000 pixels.

2. Somewhat disappointing, since rather descriptive. Readers certainly will appreciate some detailed findings (of section 3). The abstract is better in this regard.

Answer: Corrected. We tried to substantially change conclusions according to the suggestion.

Round 2

Reviewer 1 Report

The authors have satisfactorily addressed my comments and improved their manuscript in terms of clarity and completeness. I recommend its publication in Molecules.

Author Response

Thank you very much for excellent work. With compliments from the authors.